# A Joint State and Fault Estimation Scheme for State-Saturated System with Energy Harvesting Sensors

**DOI:** 10.3390/s24061967

**Published:** 2024-03-20

**Authors:** Li Zhu, Cong Huang, Quan Shi, Ruifeng Gao, Peng Ping

**Affiliations:** 1School of Transportation and Civil Engineering, Nantong University, Nantong 226019, China; 2161221@mail.dhu.edu.cn (L.Z.); grf@ntu.edu.cn (R.G.); pingpeng@ntu.edu.cn (P.P.); 2Haian Institute of High-Tech Research, Nanjing University, Nanjing 226600, China

**Keywords:** energy harvesting sensors, state and fault estimation, state saturations, time-delayed nonlinear systems, parameter uncertainties

## Abstract

In this article, the issue of joint state and fault estimation is ironed out for delayed state-saturated systems subject to energy harvesting sensors. Under the effect of energy harvesting, the sensors can harvest energy from the external environment and consume an amount of energy when transmitting measurements to the estimator. The occurrence probability of measurement loss is computed at each instant according to the probability distribution of the energy harvesting mechanism. The main objective of the addressed problem is to construct a joint state and fault estimator where the estimation error covariance is ensured in some certain sense and the estimator gain is determined to accommodate energy harvesting sensors, state saturation, as well as time delays. By virtue of a set of matrix difference equations, the derived upper bound is minimized by parameterizing the estimator gain. In addition, the performance evaluation of the designed joint estimator is conducted by analyzing the boundedness of the estimation error in the mean-squared sense. Finally, two experimental examples are employed to illustrate the feasibility of the proposed estimation scheme.

## 1. Introduction

In recent years, state estimation has gained unprecedented enthusiasm due to the swift advancement of information technology and signal processing, notably in automatic control systems, target tracking, and navigation positioning [1,2,3]. Ongoing research strives for more precise, robust, and computationally efficient state estimation methods to address the evolving demands of applications. Generally, the existing algorithms have been categorized by diverse performance criteria, including Kalman filtering and its variants, $H_{\infty }$ estimation scheme, and recursive state estimation. Specifically, the well-known Kalman filtering performs as a global optimal estimation scheme with the assumption of the statistical characteristics of the system noise being known accurately, while $H_{\infty }$ filtering relies on the infinite norm of the output-to-interference signal ratio. It should be pointed out that both the above-mentioned estimation schemes exhibit significant deviation when it comes to the nonlinearities/uncertainties. As a suboptimal Kalman-type estimation scheme, recursive state estimation is applied to accommodate the nonlinearities/uncertainties in the sense of the upper bound on estimation error covariances being minimized [4,5,6].

In existing state estimation algorithms, sensors are often assumed to operate without faults. Unfortunately, in engineering practice, various destabilizing factors emerge, including unknown external disturbances, unpredictable fluctuations in parameters, and alterations in system structure [7,8,9]. Such factors give rise to randomly occurring faults, resulting in performance degradation and/or extreme system instability. Especially in sensor networks, such faults are unmeasurable. It is essential to incorporate fault estimation, utilizing known information to estimate fault signals. To date, considerable research has been dedicated to exploring fault estimation [10,11,12,13]. For example, in [11], the issue of distributed fault-tolerant state estimation in stochastic systems across sensor networks affected by intermittent faults was explored. In [13], the investigation delved into a novel hybrid observer-based fault estimation scheme within cyber–physical systems for the joint estimation of state and faults. However, the majority of current research does not account for the impact of random faults on the results of state estimation, which constitutes one of the motivations of this article.

It is well known that measurement is one of the most crucial parts of the issue of state estimation. The conventional battery-powered approach relies on wireless sensor networks and is actually facing a series of limitations, especially the limited battery capacity [14,15]. For the sake of avoiding the energy depletion of communication devices, the so-called energy harvesting technology is invented which has the prominent feature of capturing the scattered energy resources from the external environment, encompassing solar energy, wind energy, mechanical vibration, etc. [16,17]. Then the harvested energy is converted into internally stored energy, replacing the traditional sensor power supply methods. Therefore, the wireless sensor networks achieve sustainable self-sufficiency and liberation from the limitations of sensor battery lifespans [18,19]. Nevertheless, energy harvesting sensors still bring some new challenges. Different from sensors powered by conventional batteries, a significant aspect lies in that the energy collected by the energy harvesting sensors is actually random and intermittent [20,21]. This characteristic primarily arises from the stochastic nature of the energy harvested from the external environment.

Hence, in the widespread usage of energy-harvesting sensors, ensuring continuous adequacy of stored energy at all times becomes a formidable challenge. Insufficient energy levels in sensors can lead to inevitable communication interruptions between adjacent nodes, consequently causing missing measurements. Currently, regarding the mentioned issues, energy harvesting technology has sparked a series of research interests; see, e.g., [22,23,24,25]. For instance, in [24], researchers addressed energy-dependent remote state estimation in nonlinear time-delayed systems, employing a recursive calculation for the energy level probability distribution. In [25], the state and fault estimation problem for time-varying systems with energy harvesting sensors has been investigated. In addition, inherent physical constraints in practical applications contribute to the prevalent occurrence of inevitable state saturation. Generally, state saturation is essentially the specific nonlinearity that constrains state variables in a certain boundary, which may lead to a degradation in the performance of state estimation algorithms [26,27,28]. Therefore, it is crucial to account for this phenomenon when tackling filtering/state estimation challenges. In the past decade, a substantial number of solutions and strategies have arisen for specific state estimation problems with state saturation; see, e.g., [29,30,31,32]. For example, in [30], the joint estimation problem with saturation and nonlinearity was studied, where the system was reorganized into a singular system and an estimator was designed at each node for effective estimation. In [32], a recursive filter was designed against state saturations and probabilistic attacks in an array of two-dimensional shift-varying systems.

To the best of the author’s knowledge, there has not been sufficient research on the joint estimation of state and fault with the system equipped with energy harvesting sensors, not to mention addressing the time-delayed system involving state saturation, nonlinearity, and parameter uncertainty. It is worth mentioning that most current research cannot provide performance analysis of estimation algorithms in such a complex time-delayed system. Meanwhile, existing state estimation algorithms have not fully considered the complex situations in engineering applications, making it difficult to ensure the stability of algorithm performance. As such, the main purpose of this article is to narrow such a gap by comprehensively and deeply researching a novel joint estimation scheme and establishing a sufficient condition to complete the performance analysis of the proposed algorithm. Motivated by the discussions above, the article is dedicated to investigating a joint state and fault estimation scheme for a state-saturated system subject to energy harvesting sensors. The main contributions of the article are outlined as follows:The joint state and fault estimation scheme is, for the first time, comprehensively and deeply studied for delayed nonlinear systems with energy harvesting sensors and state saturation;A set of matrix difference equations are firstly resorted to recursively compute the upper bound on the estimation error covariance (EEC) and the desired estimator gain is obtained by minimizing the trace of the upper bound on EEC;A new approach is invented to compensate for the performance degradation of the estimator caused by random faults, measurement loss due to insufficient energy level as well as sensor saturation;The mean-squared exponential boundedness of the estimation error in such a complex engineering context is first analyzed by establishing a sufficient condition.

Finally, two experimental examples are exploited to validate the feasibility of the proposed estimation algorithm.

**Notation:** The notations used in this article are standard except where otherwise stated. $A^{T}$ and $A^{-1}$ are the transposition and the inverse of *A*, respectively. For symmetric matrices *U* and *V*, U≥V(respectively, U>V), means that U−V is a positive semi-defined (positive definite) matrix. E{ξ} stands for the mathematical expectation of the stochastic variable ξ. tr{M} means the trace of the matrix *M*. Pr{χ} represents the occurrence probability of event “χ”.

## 2. Problem Formulation

The joint state and fault estimation configuration for a state-saturated system with energy harvesting sensors is shown in Figure 1, where the obtained signals are first processed through a saturation function and the sensors harvest energy for storage. The current energy level of the sensor determines whether the measurement can be transmitted. Meanwhile, the workflow of the joint estimation scheme proposed in this article can be seen in Figure 2. The system under consideration with measurements is given as follows:(1)$$\begin{array}{cc} {\vec{x}}_{s+1}= & \hbar \left(\left({\vec{A}}_{s}+\Delta {\vec{A}}_{s}\right){\vec{x}}_{s}+\left({\vec{B}}_{s}+\Delta {\vec{B}}_{s}\right){\vec{x}}_{s-\tau }+\alpha_{s}{\vec{G}}_{s}f_{s}+{\vec{D}}_{s}\omega_{s}\right)\\ {\vec{s}}_{s}= & \ell \left({\vec{C}}_{s}{\vec{x}}_{s}\right)+\nu_{s} \end{array}$$
where ${\vec{x}}_{s}\in R^{n_{x}}$ and ${\vec{y}}_{s}\in R^{n_{y}}$ are, respectively, the state and measurement output at time instant *s*. $\omega_{s}\in R^{\omega }$ and $\nu_{s}\in R^{\nu }$ are zero-mean Gaussian white-noise sequences with covariances $R_{s}>0$ and $Q_{s}>0$. τ>0 represents the constant delay. ${\vec{A}}_{s}$, ${\vec{B}}_{s}$, and ${\vec{C}}_{s}$ are known matrices with compact dimensions. In addition, $\Delta {\vec{A}}_{s}$ and $\Delta {\vec{B}}_{s}$ are parameter uncertainties with the following relationship
(2)$$\left[\begin{array}{cc} \Delta {\vec{A}}_{s} & \Delta {\vec{B}}_{s} \end{array}\right]={\vec{E}}_{s}F_{s}\left[\begin{array}{cc} {\vec{H}}_{s} & {\vec{\hat{H}}}_{s} \end{array}\right]$$
where ${\vec{E}}_{s}$, ${\vec{H}}_{s}$ and ${\vec{\hat{H}}}_{s}$ are known time-varying matrices, and $F_{s}$ denotes the parameter uncertainty satisfying $F_{s}F_{s}^{T}\leq I$.

The known nonlinear function ℓ(x) satisfies
(3)∥ℓ(ı)−ℓ(ȷ)∥ ≤ β∥ı−ȷ∥,
for all $ı,ȷ\in R^{n_{ı}}$ and a known β>0 scalar.

The saturation function ℏ(·): $R^{n_{x}}$↦$R^{n_{x}}$ is defined by
(4)$$\hbar \left(q\right)={\left[\hbar \left(q_{1}\right)\;\hbar \left(q_{2}\right)\;\cdots \;\hbar \left(q_{n_{x}}\right)\right]}^{T},$$
with $\hbar (q_{i})$ = $\mathrm{sign}\left(q_{i}\right)\;\min\{\delta_{i},|q_{i}\left|\right\}$, where $q_{i}$ and $\delta_{i}$ stand for the *i*th element of *q* and saturation level. In addition, sign(·) denotes the signum function.

The dynamic characteristics of the fault $f_{s}$ is modeled by
(5)$$f_{s+1}={𝙁}_{s}f_{s}$$
where ${𝙁}_{s}$ is a known matrix with appropriate dimension.

The variable $\alpha_{s}\in R$ is used to characterize the random nature of the fault, which satisfies the following Bernoulli distribution
(6)$$\begin{array}{c} \mathrm{Pr}\left\{\alpha_{s}=1\right\}=E\left\{\alpha_{s}\right\}=\bar{\alpha },\mathrm{Pr}\left\{\alpha_{s}=0\right\}=1-\bar{\alpha } \end{array}$$
where $\bar{\alpha }\in \left[0,1\right]$ is a known scalar.

**Remark** **1.***The purpose of this article is to design a joint estimation scheme for state and fault, which estimates randomly occurring faults while completing state estimation. The fault involved in this article is a sudden and random fault, which is very common in industrial processes. Its dynamic model is represented by Equation* (5). *When a fault signal occurs, it will be detected by the system in a timely manner, and then the fault estimation scheme will be used to estimate it. Due to the complexity of the scenarios considered in this article, accurate estimation of fault signals can improve the performance of state estimation.*

**Assumption** **A1.**
*The initial values $x_{i}$(i=1,2,…,τ) with the mean $E\left\{x_{i}\right\}≜{\bar{x}}_{i}$ and covariance $P_{i}≜E\left\{\left(x_{i}-{\bar{x}}_{i}\right){\left(x_{i}-{\bar{x}}_{i}\right)}^{T}\right\}$ are mutually uncorrelated with $\alpha_{s}$, $\omega_{s}$, and $\nu_{s}$.*


At each instant *s*, the energy level of the sensor *i* is denoted by $z_{s}^{i}\in \left\{0,1,2,\ldots ,S^{i}\right\}$, where $S^{i}$ represents the maximum storage capacity of sensor *i*. The energy harvested by the *i*th sensor at *s* is denoted by $h_{s}^{i}$, which follows an independent identically distributed random process:(7)$$\mathrm{Pr}\left\{h_{s}^{i}=j\right\}=p_{j},\;j=0,1,2,\cdots$$
with $P_{j}>0$, where $\sum_{j=0}^{\infty }p_{j}=1$ and $0\leq p_{j}\leq 1$.

When the energy harvesting sensor stores non-zero energy units, it can transmit the measurement to a remote estimator and consumes one unit of energy. Hence, the dynamics of the energy level of the sensor is denoted by
(8)$$\left\{\begin{array}{cc} z_{s+1}^{i}= & \max\{\min\left\{z_{s}^{i}+h_{s}^{i}-\Upsilon_{\left\{z_{s}^{i}>0\right\}},S^{i}\right\},0\}\\ z_{0}^{i}\leq & S^{i}, \end{array}\right.$$
and the measurement transmitted to the remote joint estimator can be written by
(9)$${\bar{y}}_{s}=\Upsilon_{\left\{z_{s}^{i}>0\right\}}{\vec{y}}_{s}$$
where $\Upsilon_{\left\{z_{s}^{i}>0\right\}}=diag\left\{\Upsilon_{1,\{z_{s}^{1}>0\}},\Upsilon_{2,\{z_{s}^{2}>0\}},\ldots ,\Upsilon_{i,\{z_{s}^{i}>0\}}\right\}$ is an indicator variable. where $\Upsilon_{i,\{z_{s}^{i}>0\}}$ can be given as
(10)$$\Upsilon_{i,\{z_{s}^{i}>0\}}=\left\{\begin{array}{c} 1,\;z_{s}^{i}>0,\\ 0,\;\mathrm{otherwise}. \end{array}\right.$$

**Remark** **2.**
*The energy harvesting sensor can be understood as a rechargeable battery that can self charge, and it can harvest, convert, and store energy through the energy harvesting mechanism. When it is necessary to transmit measurement values to a remote estimator, the sensor will evaluate its own energy level. If the energy level meets the consumption demand, the measurement transmission is completed. Otherwise, the measurement loss will emerge. Although energy harvesting sensors also have the risk of measurement loss, their loss rate is far lower than traditional power supply methods, and the impact in industrial processes is minimal.*


Setting $x_{s}=\left[{\vec{x}}_{s}^{T}\;f_{s}^{T}\right]$. Based on Equation (1), we have the following augmented system:(11)$$\left\{\begin{array}{cc} x_{s+1}= & \hbar \left(\left(A_{s}+\Delta A_{s}\right)x_{s}+\left(B_{s}+\Delta B_{s}\right)x_{s-\tau }+G_{s}x_{s}+D_{s}\omega_{s}\right)\\ y_{s}= & \ell \left(C_{s}x_{s}\right)+\nu_{s} \end{array}\right.$$
where
(12)$$\begin{array}{cc} \; & A_{s}=\left[\begin{array}{cc} {\vec{A}}_{s} & 0\\ 0 & 0 \end{array}\right],B_{s}=\left[\begin{array}{cc} {\vec{B}}_{s} & 0\\ 0 & 0 \end{array}\right],C_{s}=\left[\begin{array}{cc} {\vec{C}}_{s} & 0 \end{array}\right],D_{s}=\left[\begin{array}{c} {\vec{D}}_{s}\\ 0 \end{array}\right],G_{s}=\left[\begin{array}{cc} 0 & \alpha_{s}{\vec{G}}_{s}\\ 0 & {𝙁}_{s} \end{array}\right],\ell \left(x_{s}\right)=\left[\begin{array}{c} \ell ({\vec{x}}_{s})\\ 0 \end{array}\right],\\ \; & \left[\begin{array}{cc} \Delta A_{s} & \Delta B_{s} \end{array}\right]=E_{s}F_{s}\left[\begin{array}{cc} H_{s} & {\hat{H}}_{s} \end{array}\right],\left[\begin{array}{c} E_{s} \end{array}\right]=\left[\begin{array}{c} {\vec{E}}_{s}\\ 0 \end{array}\right],\left[\begin{array}{c} H_{s} \end{array}\right]=\left[\begin{array}{c} {\vec{H}}_{s}\\ 0 \end{array}\right],\left[\begin{array}{c} {\hat{H}}_{s} \end{array}\right]=\left[\begin{array}{c} {\vec{\hat{H}}}_{s}\\ 0 \end{array}\right]. \end{array}$$

Based on the received measurements ${\bar{y}}_{s}$, we construct the joint estimator as follows.
(13)$$\left\{\begin{array}{cc} {\hat{x}}_{s+1|s}= & \hbar \left(\left(A_{s}+G_{s}\right){\hat{x}}_{s|s}+B_{s}{\hat{x}}_{s-\tau |s-\tau }\right)\\ {\hat{x}}_{s+1|s+1}= & {\hat{x}}_{s+1|s}+K_{s+1}\left({\bar{y}}_{s+1}-\lambda_{s+1}C_{s+1}{\hat{x}}_{s+1|s}\right) \end{array}\right.$$
where ${\hat{x}}_{s+1|s}$ and ${\hat{x}}_{s+1|s+1}$ represent the one-step prediction and the estimation of $x_{s}$ at *s*, respectively. $K_{s+1}$ is the estimator gain to be designed, and $\lambda_{s+1}≜E\left\{\Upsilon_{\left\{z_{s}^{i}>0\right\}}\right\}$.

Letting the prediction error be ${\tilde{x}}_{s+1|s}≜x_{s+1}-{\hat{x}}_{s+1|s}$ and the estimation error be ${\tilde{x}}_{s+1|s+1}≜x_{s+1}-{\hat{x}}_{s+1|s+1}$, the following error dynamics are obtained from Equations (11) and (13):(14)$$\begin{array}{cc} {\tilde{x}}_{s+1|s}= & \hbar \left(\left(A_{s}+\Delta A_{s}+G_{s}\right)x_{s}+\left(B_{s}+\Delta B_{s}\right)x_{s-\tau }+D_{s}\omega_{s}\right)\\ \; & -\hbar \left(\left(A_{s}+G_{s}\right){\hat{x}}_{s|s}+B_{s}{\hat{x}}_{s-\tau |s-\tau }\right)\\ {\tilde{x}}_{s+1|s+1}= & {\tilde{x}}_{s+1|s}-K_{s+1}\left({\bar{y}}_{s+1}-\lambda_{s+1}C_{s+1}{\hat{x}}_{s+1|s}\right) \end{array}$$

The main purpose of this article is to design a joint state and fault estimator of the form Equation (13) for the considered state-saturated system Equation (1) equipped with energy harvesting sensors such that for all energy-harvesting-induced measurements and state saturations, the upper bound $\Sigma_{s+1|s+1}$ on EEC $P_{s+1|s+1}≜E\left\{{\tilde{x}}_{s+1|s+1}{\tilde{x}}_{s+1|s+1}^{T}\right\}$ is guaranteed and minimized by the estimator gain $K_{s+1}$. Furthermore, a sufficient condition is given to evaluate the boundedness analysis of the estimation error in a mean-squared sense.

## 3. Joint State and Fault Estimation

This section provides an upper bound on EEC through mathematical induction, and then such upper bound is minimized by a set of matrix difference equations. Furthermore, a sufficient criterion has been formulated to verify the exponential boundedness of the estimation error in a mean-squared sense. The following lemmas will facilitate further development of the article.

**Lemma** **1.**
*Suppose that G and H are given scalars, there exists a constant matrix $\Theta_{i}\in \left[0,\;1\right]$$\left(i=1,2,\cdots ,n_{x}\right)$ such that*

$$\hbar_{i}\left(G\right)-\hbar_{i}\left(H\right)=\Theta_{i}\left(G-H\right)$$

*where ℏ(·) stands for the saturation function in Equation (4).*


**Proof.** When G−H>0, we can have that $0\leq \hbar_{i}\left(G\right)-\hbar_{i}\left(H\right)\leq G-H$ and then we can get $\Theta_{i}=\left(\hbar_{i}\left(G\right)-\hbar_{i}\left(H\right)\right)/\left(G-H\right)\in \left[0,1\right]$. When G−H=0, we can get that $\hbar_{i}\left(G\right)-\hbar_{i}\left(H\right)=0$ and then we can choose any real number $\Theta_{i}\in \left[0,1\right]$. Similarly, when G−H<0, it is easy to complete the proof. □

**Lemma** **2.**
*Given any vectors M and N, the following inequality holds*

$$MN^{T}+NM^{T}\leq oMM^{T}+o^{-1}NN^{T}$$

*where o>0 is an arbitrary scalar.*


**Proof.** The proof can refer to [33], omitted here. □

**Lemma** **3.**
*The measurement of transmission probability at s is derived as*

$$\lambda_{s}=\mathrm{Pr}\left\{\Upsilon_{\left\{z_{s}^{i}\right\}}=1\right\}=\left[0\;\underset{S^{i}}{\underset{︸}{1\;\cdots \;1}}\right]\varphi_{s}.$$



**Proof.** The probability distribution of the energy level $z_{s}^{i}$ can be written by
$$\varphi_{s}≜{\left[\mathrm{Pr}\left\{z_{s}^{i}=0\right\}\;\mathrm{Pr}\left\{z_{s}^{i}=1\right\}\;\cdots \;\mathrm{Pr}\left\{z_{s}^{i}=S_{i}\right\}\right]}^{T}.$$Based on Equation (8), the energy $z_{s}^{i}$ is independent of $h_{s}^{i}$. Hence, for $j=0,1,\ldots ,S^{i}$, we have:
$$\begin{array}{cc} \mathrm{Pr}\{z_{s}^{i}=j\}= & \mathrm{Pr}\{\max\left\{\min\left\{z_{s}^{i}+h_{s}^{i}-\Upsilon_{\left\{z_{s}^{i}>0\right\}},S^{i}\right\},0\right\}=j\}\\ = & \mathrm{Pr}\left\{z_{s}^{i}=0,h_{s}^{i}=j\right\}+\sum_{\iota =1}^{j+1}\mathrm{Pr}\left\{z_{s}^{i}=\iota ,h_{s}^{i}=j+1-\iota \right\}\\ = & \mathrm{Pr}\left\{z_{s}^{i}=0\right\}\mathrm{Pr}\left\{h_{s}^{i}=j\right\}+\sum_{\iota =1}^{j+1}\mathrm{Pr}\left\{z_{s}^{i}=\iota \right\}\mathrm{Pr}\left\{h_{s}^{i}=j+1-\iota \right\}\\ = & \mathrm{Pr}\left\{z_{s}^{i}=0\right\}p_{j}+\sum_{\iota =1}^{j+1}\mathrm{Pr}\left\{z_{s}^{i}=\iota \right\}p_{j+1-\iota }. \end{array}$$Then, the probability $\mathrm{Pr}\{z_{s}^{i}=S^{i}\}$ can be expressed by
$$\begin{array}{cc} \mathrm{Pr}\{z_{s}^{i}=S^{i}\}= & 1-\sum_{j=1}^{S^{i}-1}\mathrm{Pr}\left\{z_{s}^{i}=j\right\}=1-\mathrm{Pr}\left\{z_{s}^{i}=0\right\}\sum_{j=1}^{S^{i}-1}p_{j}-\sum_{j=1}^{S^{i}-1}\sum_{\iota =1}^{j+1}\mathrm{Pr}\left\{z_{s}^{i}=\iota \right\}p_{j+1-\iota }. \end{array}$$Here, for the energy level ${\left\{z_{s}^{i}\right\}}_{s\geq 0}$ in Equation (8), the recursion of the probability distribution $\varphi_{s}$ can be computed by
$$\left\{\begin{array}{cc} \varphi_{s+1}= & \varsigma +\Phi \varphi_{s}\\ \varphi_{0}= & \left[\underset{z_{0}^{i}}{\underset{︸}{0\;\cdots \;0}}\;1\;\underset{S^{i}-z_{0}^{i}}{\underset{︸}{0\;\cdots \;0}}\right] \end{array}\right.$$
where $\varsigma ={\left[{\underset{︸}{0\;\cdots \;0}}_{S^{i}}\;1\right]}^{T}$ and
$$\Phi =\left[\begin{array}{ccccc} p_{0} & p_{0} & 0 & \cdots & 0\\ p_{1} & p_{1} & p_{0} & \cdots & 0\\ p_{2} & p_{2} & p_{1} & \cdots & 0\\ \vdots & \vdots & \vdots & \ddots & \vdots \\ p_{S^{i}-1} & p_{S^{i}-1} & p_{S^{i}-2} & \cdots & p_{0}\\ -\sum_{j=0}^{S^{i}-1}p_{j} & -\sum_{j=0}^{S^{i}-1}p_{j} & -\sum_{j=0}^{S^{i}-2}p_{j} & \cdots & -p_{0} \end{array}\right].$$According to Equation (10), one immediately has
$$\begin{array}{cc} \mathrm{Pr}\{\Upsilon_{\left\{z_{s}^{i}\right\}}=0\}= & \mathrm{Pr}\left\{\left\{z_{s}^{i}\right\}=0\right\}=\left[1\;\underset{S^{i}}{\underset{︸}{0\;\cdots \;0}}\right]\varphi_{s},\\ \mathrm{Pr}\{\Upsilon_{\left\{z_{s}^{i}\right\}}=1\}= & 1-\mathrm{Pr}\left\{\left\{z_{s}^{i}\right\}=0\right\}=\left[0\;\underset{S^{i}}{\underset{︸}{1\;\cdots \;1}}\right]\varphi_{s}, \end{array}$$
which ends the proof. □

With the help of Lemma 1, Equation (14) can be rewritten as
(15)$$\begin{array}{cc} {\tilde{x}}_{s+1|s}= & \Theta_{s}\left(\left(A_{s}+G_{s}\right){\tilde{x}}_{s|s}+\Delta A_{s}x_{s}+B_{s}{\tilde{x}}_{s-\tau |s-\tau }+\Delta B_{s}x_{s-\tau }+D_{s}\omega_{s}\right)\\ {\tilde{x}}_{s+1|s+1}= & {\tilde{x}}_{s+1|s}-K_{s+1}\left({\bar{y}}_{s+1}-\lambda_{s+1}C_{s+1}{\hat{x}}_{s+1|s}\right)\\ = & \left(I-\lambda_{s+1}K_{s+1}C_{s+1}\right){\tilde{x}}_{s+1|s}-\Upsilon_{\left\{z_{s+1}^{i}>0\right\}}K_{s+1}\ell \left(C_{s+1}x_{s+1}\right)\\ \; & +\lambda_{s+1}K_{s+1}C_{s+1}x_{s+1}-\Upsilon_{\left\{z_{s+1}^{i}>0\right\}}K_{s+1}\nu_{s+1} \end{array}$$
where $\Theta_{s}≜diag\left\{\theta_{1,s},\theta_{2,s},\cdots ,\theta_{i,s}\right\}$ with $\theta_{i,s}\in \left[0,\;1\right]$(i=1,2,⋯,m).

By noting $P_{s+1|s}≜E\left\{{\tilde{x}}_{s+1|s}{\tilde{x}}_{s+1|s}^{T}\right\}$ and $P_{s+1|s+1}≜E\left\{{\tilde{x}}_{s+1|s+1}{\tilde{x}}_{s+1|s+1}^{T}\right\}$, the prediction error covariance and EEC are, respectively, given by
(16)$$\begin{array}{cc} \; & P_{s+1|s}\\ = & E\left\{\Theta_{s}\left(\left(A_{s}+G_{s}\right){\tilde{x}}_{s|s}+\Delta A_{s}x_{s}+B_{s}{\tilde{x}}_{s-\tau |s-\tau }+\Delta B_{s}x_{s-\tau }+D_{s}\omega_{s}\right)\right.\\ \; & \times \left.{\left(\left(A_{s}+G_{s}\right){\tilde{x}}_{s|s}+\Delta A_{s}x_{s}+B_{s}{\tilde{x}}_{s-\tau |s-\tau }+\Delta B_{s}x_{s-\tau }+D_{s}\omega_{s}\right)}^{T}\Theta_{s}^{T}\right\}, \end{array}$$
and
(17)$$\begin{array}{cc} \; & P_{s+1|s+1}\\ = & E\left\{\left(I-\lambda_{s+1}K_{s+1}C_{s+1}\right){\tilde{x}}_{s+1|s}{\tilde{x}}_{s+1|s}^{T}\left(I-\lambda_{s+1}K_{s+1}C_{s+1}\right.{)}^{T}-\left(I-\lambda_{s+1}K_{s+1}C_{s+1}\right)\right.\\ \; & \times \left.{\tilde{x}}_{s+1|s}\Upsilon_{\left\{z_{s+1}^{i}>0\right\}}\ell^{T}\left(x_{s+1}C_{s+1}\right)K_{s+1}^{T}+\left(I-\lambda_{s+1}K_{s+1}C_{s+1}\right){\tilde{x}}_{s+1|s}x_{s+1}^{T}C_{s+1}^{T}K_{s+1}^{T}\right.\\ \; & \times \left.\lambda_{s+1}^{T}-\Upsilon_{\left\{z_{s+1}^{i}>0\right\}}K_{s+1}\ell \left(x_{s+1}C_{s+1}\right){\tilde{x}}_{s+1|s}^{T}{\left(I-\lambda_{s+1}K_{s+1}C_{s+1}\right)}^{T}+\Upsilon_{\left\{z_{s+1}^{i}>0\right\}}K_{s+1}\right.\\ \; & \times \left.\ell \left(x_{s+1}C_{s+1}\right)\Upsilon_{\left\{z_{s+1}^{i}>0\right\}}\ell^{T}\left(x_{s+1}C_{s+1}\right)K_{s+1}^{T}-\Upsilon_{\left\{z_{s+1}^{i}>0\right\}}K_{s+1}\ell \left(x_{s+1}C_{s+1}\right)x_{s+1}^{T}C_{s+1}^{T}\right.\\ \; & \times \left.K_{s+1}^{T}\lambda_{s+1}^{T}+\lambda_{s+1}K_{s+1}C_{s+1}x_{s+1}{\tilde{x}}_{s+1|s}^{T}{\left(I-\lambda_{s+1}K_{s+1}C_{s+1}\right)}^{T}-\lambda_{s+1}K_{s+1}C_{s+1}\right.\\ \; & \times \left.x_{s+1}\Upsilon_{\left\{z_{s+1}^{i}>0\right\}}\ell^{T}\left(x_{s+1}C_{s+1}\right)K_{s+1}^{T}+\lambda_{s+1}K_{s+1}C_{s+1}x_{s+1}x_{s+1}^{T}C_{s+1}^{T}K_{s+1}^{T}\lambda_{s+1}^{T}\right.\\ \; & +\left.\Upsilon_{\left\{z_{s+1}^{i}>0\right\}}K_{s+1}\nu_{s+1}\nu_{s+1}^{T}K_{s+1}^{T}\Upsilon_{\left\{z_{s+1}^{i}>0\right\}}\right\}. \end{array}$$

**Theorem** **1.***Let the positive scalars*$\varepsilon_{i,s}\left(i=1,2,\cdots ,m\right)$ *be given. The prediction error covariance and EEC are given in Equations* (16) *and* (17)*, respectively. If the matrices satisfy the following equations*
(18)$$\begin{array}{cc} \Sigma_{s+1|s}≜ & \min\left\{\Omega_{s}I+D_{s}R_{s}D_{s}^{T},4\rho I\right\}, \end{array}$$*and*
(19)$$\begin{array}{cc} \Sigma_{s+1|s+1} & ≜\left(1+\varepsilon_{9,s}+\varepsilon_{10,s}\right)\left(I-\lambda_{s+1}K_{s+1}C_{s+1}\right)\Sigma_{s+1|s}{\left(I-\lambda_{s+1}K_{s+1}C_{s+1}\right)}^{T}\\ \; & +\left(1+\varepsilon_{9,s}^{-1}+\varepsilon_{11,s}\right)\beta^{2}\lambda_{s+1}K_{s+1}C_{s+1}W_{s+1}C_{s+1}^{T}K_{s+1}^{T}+\left(1+\varepsilon_{10,s}^{-1}+\varepsilon_{11,s}^{-1}\right)\\ \; & \times \lambda_{s+1}K_{s+1}C_{s+1}W_{s+1}C_{s+1}^{T}K_{s+1}^{T}\lambda_{s+1}^{T}+\lambda_{s+1}K_{s+1}Q_{s+1}K_{s+1}^{T} \end{array}$$*with*
$P_{0|0}\leq \Sigma_{0|0}$*, where*

$$\begin{array}{cc} \Omega_{s} & ≜\mathrm{tr}\left\{\left(1+\varepsilon_{1,s}+\varepsilon_{2,s}+\varepsilon_{3,s}\right)\left(A_{k}+G_{s}\right)\Sigma_{s|s}{\left(A_{s}+G_{s}\right)}^{T}+\left(1+\varepsilon_{1,s}^{-1}+\varepsilon_{4,s}+\varepsilon_{5,s}\right)\right.\\ \; & \times \left.W_{s}\mathrm{tr}\left\{H_{s}H_{s}^{T}\right\}E_{k}E_{s}^{T}+\left(1+\varepsilon_{2,s}^{-1}+\varepsilon_{4,s}^{-1}+\varepsilon_{6,s}\right)B_{s}\Sigma_{s-\tau |s-\tau }B_{s}^{T}\right.\\ \; & +\left.\left(1+\varepsilon_{3,s}^{-1}+\varepsilon_{5,s}^{-1}+\varepsilon_{6,s}^{-1}\right)V_{s}\mathrm{tr}\left\{{\hat{H}}_{s}{\hat{H}}_{s}^{T}\right\}E_{s}E_{s}^{T}\right\},\rho ≜\sum_{i=0}^{n}\delta_{i}^{2},\\ W_{s} & ≜\min\left\{\mathrm{tr}\left\{\left(1+\varepsilon_{7,s}\right)\Sigma_{s|s-1}+\left(1+\varepsilon_{7,s}^{-1}\right)E\left\{{\hat{x}}_{s|s-1}{\hat{x}}_{s|s-1}^{T}\right\}\right\},\rho I\right\},\\ V_{s} & ≜\min\left\{\mathrm{tr}\left\{\left(1+\varepsilon_{8,s}\right)\Sigma_{s-\tau |s-1-\tau }+\left(1+\varepsilon_{8,s}^{-1}\right)E\left\{{\hat{x}}_{s-\tau |s-1-\tau }{\hat{x}}_{s-\tau |s-1-\tau }^{T}\right\}\right\},\rho I\right\}, \end{array}$$

*then $\Sigma_{s+1|s+1}$ is an upper bound on $P_{s+1|s+1}$, i.e., $P_{s+1|s+1}\leq \Sigma_{s+1|s+1}$.*
**Proof.** The proof of Theorem 1 is given in Appendix A. □

In the following theorem, the joint estimator gain $K_{s}$ has been design to minimize the upper bound in Theorem 1.

**Theorem** **2.**
*The trace of the upper bound on EEC in Theorem 1 is minimized by the following estimator gain*

(20)
$$K_{s+1}={ℵ}_{s+1}\Pi_{s+1}^{-1}$$

*where*

$$\begin{array}{cc} {ℵ}_{s+1}≜ & \left(1+\varepsilon_{9,s}+\varepsilon_{10,s}\right)\lambda_{s+1}\Sigma_{s+1|s}C_{s+1}^{T},\\ \Pi_{s+1}≜ & \left(1+\varepsilon_{9,s}+\varepsilon_{10,s}\right)\lambda_{s+1}^{2}C_{s+1}\Sigma_{s+1|s}C_{s+1}^{T}+\left(1+\varepsilon_{9,s}^{-1}+\varepsilon_{11,s}\right)\beta^{2}\lambda_{s+1}C_{s+1}W_{s+1}C_{s+1}^{T}\\ \; & +\left(1+\varepsilon_{10,s}^{-1}+\varepsilon_{11,s}^{-1}\right)\lambda_{s+1}C_{s+1}W_{s+1}C_{s+1}^{T}\lambda_{s+1}^{T}+\lambda_{s+1}Q_{s+1}. \end{array}$$



**Proof.** The trace of the $\Sigma_{s+1|s+1}$ can be computed as follows:
(21)$$\begin{array}{cc} \; & \mathrm{tr}\{\Sigma_{s+1|s+1}\}\\ = & \left(1+\varepsilon_{9,s}+\varepsilon_{10,s}\right)\mathrm{tr}\left\{\left(I-\lambda_{s+1}K_{s+1}C_{s+1}\right)\Sigma_{s+1|s}{\left(I-\lambda_{s+1}K_{s+1}C_{s+1}\right)}^{T}\right\}\\ \; & +\left(1+\varepsilon_{9,s}^{-1}+\varepsilon_{11,s}\right)\beta^{2}\lambda_{s+1}\mathrm{tr}\left\{K_{s+1}C_{s+1}\Sigma_{s+1|s}C_{s+1}^{T}K_{s+1}^{T}\right\}+\left(1+\varepsilon_{10,s}^{-1}+\varepsilon_{11,s}^{-1}\right)\lambda_{s+1}\\ \; & \times \mathrm{tr}\left\{K_{s+1}C_{s+1}W_{s+1}C_{s+1}^{T}K_{s+1}^{T}\right\}+\lambda_{s+1}\mathrm{tr}\left\{K_{s+1}Q_{s+1}K_{s+1}^{T}\right\}. \end{array}$$In order to compute the optimal estimator gain $K_{s+1}$, we take the partial derivative of the trace of $\Sigma_{s+1|s+1}$ with respect to $K_{s+1}$
(22)$$\begin{array}{cc} \; & \frac{\partial }{\partial K_{s+1}}\mathrm{tr}\left\{\Sigma_{s+1|s+1}\right\}\\ = & -2\left(1+\varepsilon_{9,s}+\varepsilon_{10,s}\right)\lambda_{s+1}\Sigma_{s+1|s}C_{s+1}^{T}+2\left(1+\varepsilon_{9,s}+\varepsilon_{10,s}\right)\lambda_{s+1}^{2}K_{s+1}C_{s+1}\Sigma_{s+1|s}C_{s+1}^{T}\\ \; & +2\left(1+\varepsilon_{9,s}^{-1}+\varepsilon_{11,s}\right)\beta^{2}\lambda_{s+1}K_{s+1}C_{s+1}W_{s+1|k}C_{s+1}^{T}+2\left(1+\varepsilon_{10,s}^{-1}+\varepsilon_{11,s}^{-1}\right)\lambda_{s+1}K_{s+1}\\ \; & \times C_{s+1}W_{s+1|s}C_{s+1}^{T}+2\lambda_{s+1}K_{s+1}Q_{s+1}. \end{array}$$Letting $\frac{\partial }{\partial K_{s+1}}\mathrm{tr}\left\{\Sigma_{s+1|s+1}\right\}=0$, one has
(23)$$K_{s+1}={ℵ}_{s+1}\Pi_{s+1}^{-1},$$
which ends the proof. □

**Remark** **3.**
*Based on the above discussion, we obtained an upper bound on the EEC in Theorem 1, and in Theorem 2, we obtained the estimator gain by minimizing this upper bound. Therefore, we have successfully resolved the issue of the joint state and fault estimation for the state-saturated system equipped with energy harvesting sensors. In addition, to better demonstrate the workflow of our algorithm, a simple flowchart is shown in Figure 2.*


In the following theorem, we are poised to assess the effectiveness of the designed estimation scheme and establish a sufficient condition to ensure the exponential boundedness of the estimation error in a mean-squared sense.

**Definition** **1.**
*The stochastic process $\xi_{s}$ is considered to be exponentially bounded in mean-squared sense if there exist real numbers a>0,b>0 and 0<c<1 such that*

$$E\left\{∥\xi_{s}{∥}^{2}\right\}\leq aE\left\{∥\xi_{0}{∥}^{2}\right\}b^{s}+c$$

*holds for any s≥0.*


**Theorem** **3.**
*Suppose that $\bar{a}$, $\bar{b}$, $\underline{b}$, $\bar{c}$, $\underline{c}$, $\bar{d}$, $\underline{d}$, $\bar{e}$, $\bar{f}$, $\bar{g}$, $\bar{h}$, $\bar{\hat{h}}$, $\bar{\lambda }$, $\underline{\lambda }$, $\bar{\theta }$, ${\bar{\vartheta }}_{1}$, ${\bar{\vartheta }}_{2}$, ${\bar{\vartheta }}_{3}$, ${\bar{\vartheta }}_{4}$, $\bar{r}$, $\bar{s}$, $\bar{\chi }$, $\bar{\omega }$, $\underline{\omega }$, $\bar{\nu }$, $\bar{k}$, $\bar{\psi }$, $\underline{\psi }$ are positive scalars, if the following inequalities*

(24)
$$\begin{array}{cc} \; & \|A_{s}\|\leq \bar{a},\underline{b}\leq \|B_{s}\|\leq \bar{b},\underline{c}\leq \|C_{s}\|\leq \bar{c},\underline{d}\leq \|D_{s}\|\leq \bar{d},\|E_{s}\|\leq \bar{e},\|F_{s}\|\leq \bar{f},\left\|G_{s}\right\|\leq \bar{g},\\ \; & \|H_{s}\|\leq \bar{h},\|{\hat{H}}_{s}\|\leq \bar{\hat{h}},\underline{\omega }\leq \|R_{s}\|\leq \bar{\omega },\|Q_{s}\|\leq \bar{\nu },\|\Theta_{s}\|\leq \bar{\theta },\underline{\lambda }\leq \left\|\lambda_{s}\right\|\leq \bar{\lambda },\\ \; & \mathrm{tr}\left\{E\left\{x_{s-\tau |s-\tau }x_{s-\tau |s-\tau }^{T}\right\}\right\}\leq {\bar{\vartheta }}_{1},\mathrm{tr}\left\{E\left\{x_{s}x_{s}^{T}\right\}\right\}\leq {\bar{\vartheta }}_{2},\mathrm{tr}\left\{E\left\{x_{s-\tau }x_{s-\tau }^{T}\right\}\right\}\leq {\bar{\vartheta }}_{3},\\ \; & \mathrm{tr}\left\{E\left\{\ell \left(C_{s}x_{s}\right)\ell^{T}\left(C_{s}x_{s}\right)\right\}\right\}\leq {\bar{\vartheta }}_{4},\varrho ={\left(1+\frac{{\bar{\lambda }}^{2}{\bar{c}}^{2}}{{\underline{\lambda }}^{2}{\underline{c}}^{2}}\right)}^{2}{\bar{\theta }}^{2}{\left(\bar{a}+\bar{g}\right)}^{2}<1, \end{array}$$

*hold, then the estimation error is exponentially bounded in mean-squared sense.*


**Proof.** The proof of Theorem 3 is given in Appendix B. □

## 4. Simulation Experiments

In this section, we intend to provide two experimental examples to show the feasibility of the proposed joint state and fault estimation scheme. Moreover, in Example 1, a performance comparison was made between the Kalman filtering and the proposed algorithm.

**Example** **1.***Consider the delayed state-saturated system Equation* (1) *equipped with energy harvesting sensors with the following parameters:*
$$\begin{array}{cc} \; & {\vec{A}}_{s}=\left[\begin{array}{cc} 0.39 & -0.01+0.15*\cos(s)\\ 0.15 & 0.38 \end{array}\right],{\vec{B}}_{s}=\left[\begin{array}{cc} 0.2 & 0\\ 0 & 0.15 \end{array}\right],{\vec{H}}_{s}=\left[\begin{array}{cc} 0.01*\cos(s) & 0.02 \end{array}\right],\\ \; & {\vec{\hat{H}}}_{s}=\left[\begin{array}{cc} 0.01 & 0.05*\sin(s) \end{array}\right],{\vec{C}}_{s}=\left[\begin{array}{cc} 2.5 & 0.2+0.15*\cos(s) \end{array}\right],F_{s}=\sin\left(s\right),{𝙁}_{s}=2\sin\left(s\right),\\ \; & \ell \left(x\right)=0.17\sin\left(x\right),{\vec{D}}_{s}=\left[\begin{array}{c} 0.15\\ 0.18 \end{array}\right],{\vec{E}}_{s}=\left[\begin{array}{c} 0.1\\ 0.1 \end{array}\right],{\vec{G}}_{s}=\left[\begin{array}{c} 0\\ 0.01 \end{array}\right]. \end{array}$$*The noise $\omega_{s}$ and $\nu_{s}$ are zero-mean Gaussian noises, respectively, with covariances $R_{s}=0.2$ and $Q_{s}=0.1$. The time delay is set as τ=1 and the saturation level is set as $\delta_{1}=\delta_{2}=0.1$. Suppose that the initial energy unit stored in the sensor is $z_{0}=1$, meanwhile, the sensor has a maximum storage capacity of S=3 energy units. The random variable is chosen as $\bar{\alpha }=0.95$. Based on the above parameters, the estimator parameter $K_{s+1}$ can be computed at each instant by recurring to Equation* (20)*. Furthermore, the recursive computation of the probability distribution of the sensor energy level $\varphi_{s}$ and the expectation of the measurement transmission $\lambda_{s}$ is presented in Table 1.*
*The main results are listed in Figure 3, Figure 4, Figure 5 and Figure 6. Figure 3 plots the actual states and their estimates for $x_{s}^{1}$ and $x_{s}^{2}$. Figure 5 depicts the trace of the minimum upper bound $\Sigma_{s|s}$ and the mean square error (MSE) (defined by $MSE_{s}≜\frac{1}{300}\sum_{t=1}^{300}\sum_{l=1}^{2}\left(x_{s}^{l}-{\hat{x}}_{s|s}^{l}\right){\left(x_{s}^{l}-{\hat{x}}_{s|s}^{l}\right)}^{T}$) for the estimation of the state variables. The faults and their estimates are depicted in Figure 4. Figure 6 depicts the values of $h_{s}$ and $\Upsilon_{\left\{z_{s}>0\right\}}$ at each instant. Overall, based on the comparison results between the proposed algorithm and the Kalman filtering algorithm shown in Figure 3, Figure 4 and Figure 5, it is not difficult to find that the proposed algorithm has better estimation performance when facing the complex situations considered in this article.*


**Example** **2.***Considering the target tracking task [34] in Equation* (1) *with the following parameters:*
$$\begin{array}{cc} \; & {\vec{A}}_{s}=\left[\begin{array}{cccc} 1 & 0 & \Delta t & 0\\ 0 & 1 & 0 & \Delta t\\ 0 & 0 & 1 & 0\\ 0 & 0 & 0 & 1 \end{array}\right],{\vec{B}}_{s}=\left[\begin{array}{cccc} 0.01 & 0 & 0 & 0\\ 0 & 0.01 & 0 & 0\\ 0 & 0 & 0.01 & 0\\ 0 & 0 & 0 & 0.01 \end{array}\right],\\ \; & {\vec{H}}_{s}=\left[\begin{array}{cccc} 0.01\cos(s) & 0.01 & 0.01 & 0.01 \end{array}\right],{\vec{\hat{H}}}_{s}=\left[\begin{array}{cccc} 0.01 & 0.01\sin(s) & 0.01 & 0.01 \end{array}\right],\\ \; & {\vec{C}}_{s}=\left[\begin{array}{cccc} 0.01 & 0.01 & 0.01 & 0.01 \end{array}\right],\ell \left(x_{s}\right)=0.8\sin\left(s\right),\\ \; & {\vec{D}}_{k}=\left[\begin{array}{c} 0.01\\ 0.01\\ 0.01\\ 0.01 \end{array}\right],{\vec{E}}_{s}=\left[\begin{array}{c} 0.01\\ 0.01\\ 0.01\\ 0.01 \end{array}\right],{\vec{G}}_{s}=\left[\begin{array}{cccc} 0.01 & 0 & 0 & 0\\ 0 & 0.01 & 0 & 0\\ 0 & 0 & 0 & 0\\ 0 & 0 & 0 & 0 \end{array}\right]. \end{array}$$
*The initial state of the target system is given as $x_{0}=\left[\begin{array}{cccc} 0 & 0 & 0.1m/s & 0.1m/s \end{array}\right]$. We set $R_{s}=0.01$ and $Q_{s}=0.01$, and we choose the saturation levels as $\delta_{1}=\delta_{2}=50$. The sampling period is selected as Δt=1. Other parameters are the same as Example 1.*

*The simulation results are shown in Figure 7 and Figure 8. To be specific, Figure 7 plots the actual moving trajectory and the position estimate of the target in the two-dimensional plane. Moreover, the plotted figures in Figure 8 illustrate both the actual coordinates $x_{s}^{1}$ and $x_{s}^{2}$ of the target position and their corresponding estimates. It is evident that the proposed estimation scheme exhibits satisfactory performance.*


## 5. Conclusions

In this article, we have committed to studying a joint state and fault estimation scheme for state-saturated system subject to energy harvesting sensors. The energy-harvesting-induced missing measurements have been considered where the occurrence probability of this phenomenon has been computed at each instant. A joint state and fault estimator has been developed where upper bound on EEC has been ensured and then be minimized by appropriately designing the joint estimator gain. Moreover, the performance of the estimation scheme has been conducted by analyzing the boundedness of the estimation error. Finally, two experimental examples are employed to illustrate the effectiveness of the proposed estimation scheme, and the results show that the designed estimator has excellent estimation performance and performs well in target tracking. With the continuous expansion of application scenarios, state estimation/filtering algorithms are facing increasingly unstable factors, and relevant research should be based on industrial applications to make algorithms adapt to actual demands. In the future, based on the research results of this article, we will devote to the state estimation problems subject to time-correlated fading channels [35] and multiple description coding scheme [36].

## Figures and Tables

**Figure 1 sensors-24-01967-f001:**
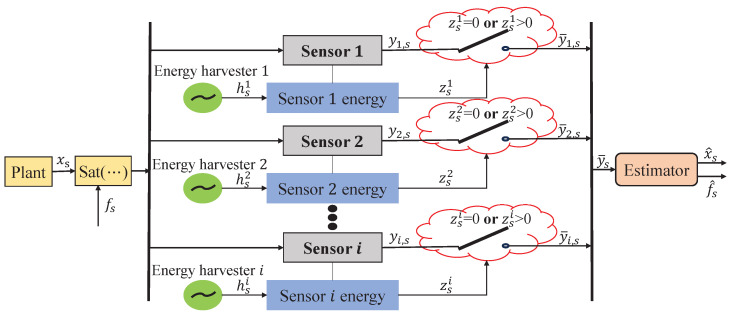
The joint state and fault estimation configuration for state-saturated system with energy harvesting sensors.

**Figure 2 sensors-24-01967-f002:**
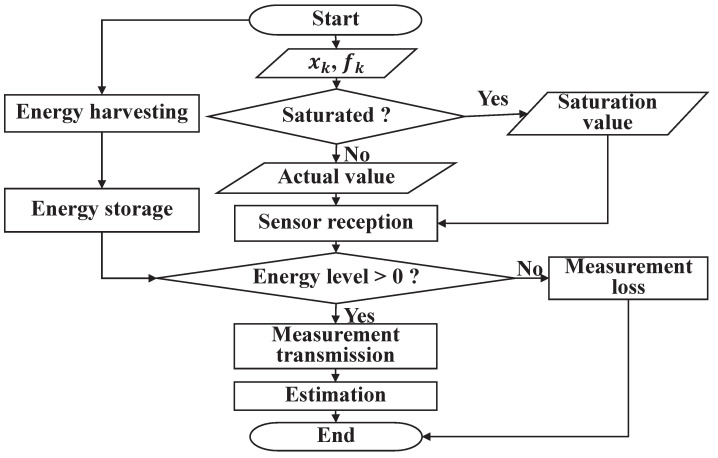
The workflow diagram of the joint state and fault estimation scheme.

**Figure 3 sensors-24-01967-f003:**
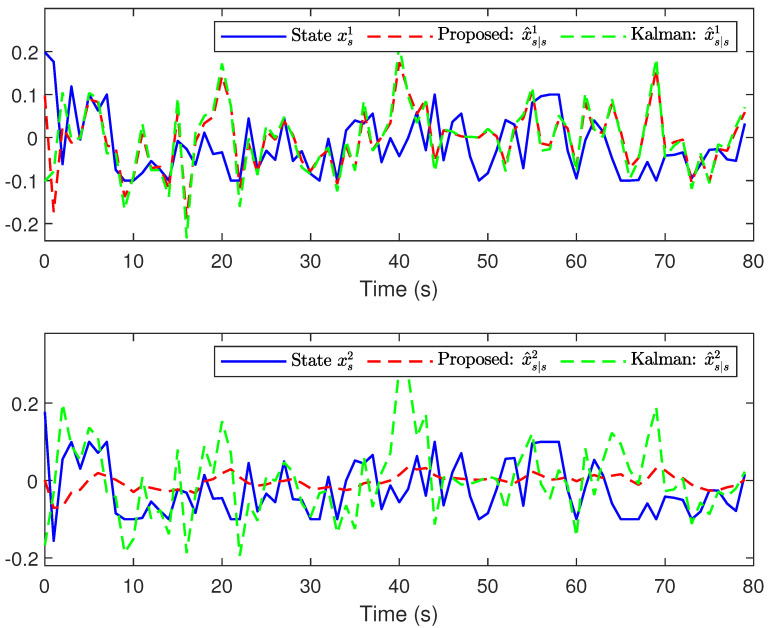
State $x_{s}^{1}$,$x_{s}^{2}$ and their estimates.

**Figure 4 sensors-24-01967-f004:**
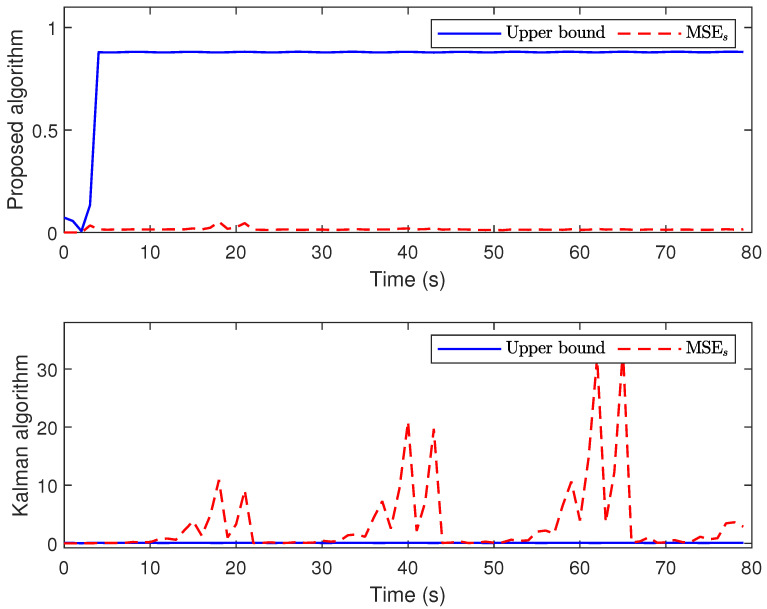
The trace of the upper bounds and estimation error covariances.

**Figure 5 sensors-24-01967-f005:**
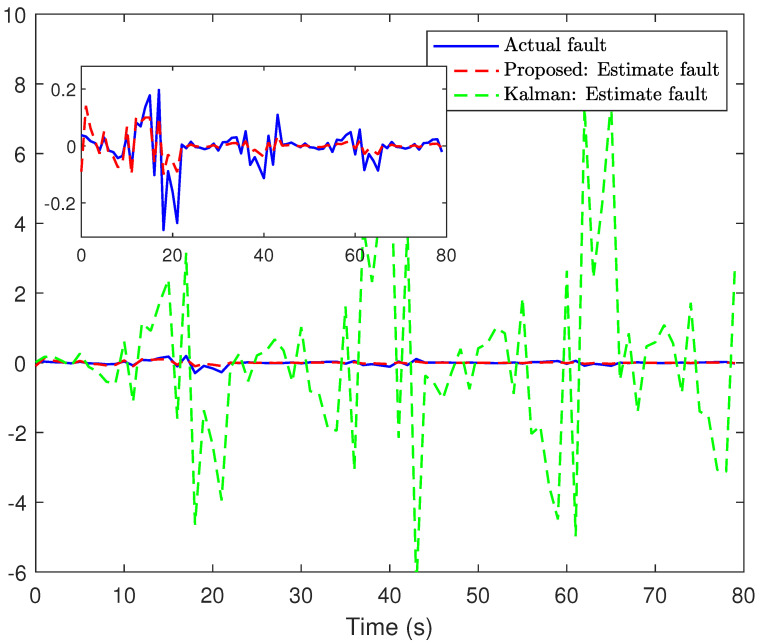
The faults and their estimates.

**Figure 6 sensors-24-01967-f006:**
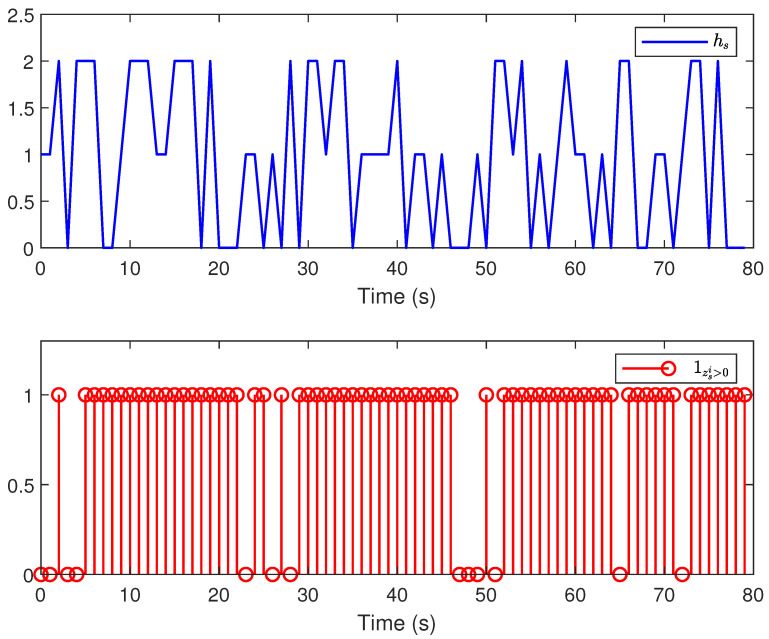
The energy harvested and energy consumption at time instant *s*.

**Figure 7 sensors-24-01967-f007:**
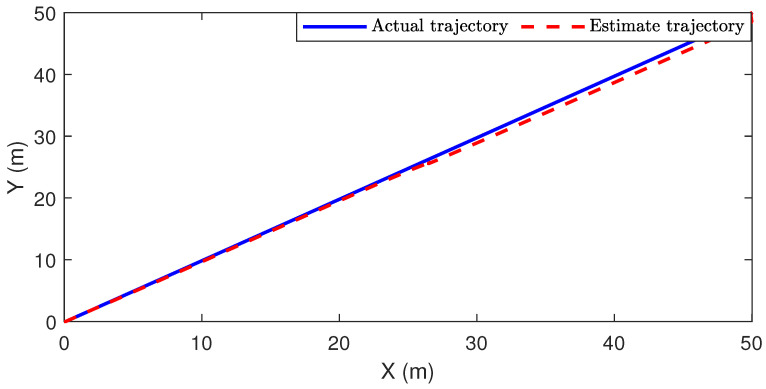
The actual moving trajectory and the position estimate of the target.

**Figure 8 sensors-24-01967-f008:**
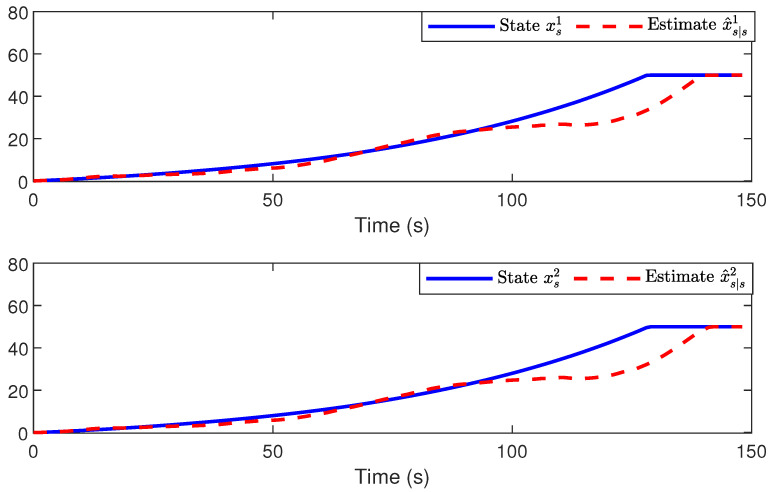
The actual coordinates $x_{s}^{1}$ and $x_{s}^{2}$ of the target position and their estimates.

**Table 1 sensors-24-01967-t001:** Results for energy harvesting sensors.

*s*	0	1	2	3	⋯
$\lambda_{s}$	1	0.8750	0.8875	0.9125	⋯
$\varphi_{s}$	$\left[\begin{array}{c} 0\\ 1\\ 0\\ 0 \end{array}\right]$	$\left[\begin{array}{c} 0.1250\\ 0.2125\\ 0.2875\\ 0.3750 \end{array}\right]$	$\left[\begin{array}{c} 0.1125\\ 0.2125\\ 0.3000\\ 0.3750 \end{array}\right]$	$\left[\begin{array}{c} 0.0875\\ 0.2375\\ 0.3000\\ 0.3750 \end{array}\right]$	⋯

## Data Availability

No new data were created or analyzed in this study. Data sharing is not applicable to this article.

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
