# Peer review of "A Joint State and Fault Estimation Scheme for State-Saturated System with Energy Harvesting Sensors"

_sensors, 2024, doi:10.3390/s24061967_

Round 1

Reviewer 1 Report

Comments and Suggestions for Authors

Good day!

The authors need to specify the object of research and its practical application. Such a detailed mathematical apparatus is difficult for perception and readers. I recommend to simplify the mathematical apparatus, avoid formulas to the floor floor. Moreover, most variables are not disclosed under the formula! Requires refinement ...

It is required to reveal the definition of "Energy Harvesting Sensors", it is not clear what it is ...

The annotation talks about two experimental examples proving the work ... The mathematical model is hardly an experiment.

Conclusions do not reflect the results declared in the annotation and obtained during the work

Reviewer 2 Report

Comments and Suggestions for Authors

This paper develops a joint state and fault estimation approach for systems with energy harvesting sensors, addressing challenges like energy constraints and time delays, and validates its effectiveness through experiments. Several major concerns and specific comments are listed below:

1.      The numbering of references in your manuscript is perplexing. I suggest adopting a sequential numbering format, such as 1, 2, 3, 4, for clarity and ease of understanding. Meanwhile, The literature review section should provide a more detailed introduction to the current state of research.

2.      The references should be sourced from the Web of Science, specifically covering the period from 2021 to 2024. It is essential that at least 50% of all references, with a minimum of 20 references.

3.      The title of Chapter 3 requires revision. Additionally, consider relocating the extensive formula derivations found within this chapter to an appendix instead of including them in the main text.

4.      It's unclear what "Fault Estimation" refers to in your manuscript. The term "fault" is ambiguous. Furthermore, Figure 4 appears to illustrate Remaining Useful Life rather than fault estimation. Please clarify these points.

5.      The "contributions" section must emphasize the theoretical or practical impacts of the work, going beyond simple summarization. It should clearly delineate how this research advances the field, particularly in relation to cutting-edge methodologies like the one discussed in the recent publication such as Reliability Engineering & System Safety, 10.1016/j.ress.2024.110014(A Hybrid Prognosis Scheme for Rolling Bearings Based on a Novel Health Indicator and Nonlinear Wiener Process), authors need to do a better job of highlighting their contributions.

6.      The paper must include a thorough discussion on the limitations and shortcomings of the model proposed.

7.      The final sentence of the conclusion is ambiguous and requires clarification. What is the intended meaning?

8.      When referring to an equation number in the text, it should be formatted as "Eq. (17)" or "Equation 17".

9.      A flowchart illustrating the workflow of this paper is recommended.

Comments on the Quality of English Language

The quality of English expression needs improvement; many sections seem to be direct translations from Chinese, impacting the clarity and fluency of the text.

Reviewer 3 Report

Comments and Suggestions for Authors

Comments to the authors:

1. The authors should cite references follows the order given in the reference list. Particularly, the first reference cited must be reference [1] (not reference [3])

2. Main difference between this paper and the other published works should be presented more clearly.

3. The authors should add more explanation/discussion for Figure 1.

4. The authors should provide proves for Lammas in this paper.

5. Performance comparison with the existing works can be added in this paper

6. Conclusion section should be improved with more recommendation/useful designs.

7. There are several minor typos and grammatical errors in this paper.   

Comments on the Quality of English Language

Minor editing of English language is required

Round 2

Reviewer 1 Report

Comments and Suggestions for Authors

no comments yet

Reviewer 2 Report

Comments and Suggestions for Authors

This paper has significantly improved in quality through revisions and I recommend accepting it.

Reviewer 3 Report

Comments and Suggestions for Authors

The Reviewer have no further comment.

Comments on the Quality of English Language

 Minor editing of English language is required.